# Outcome Impacts Due to Pathogen-Specific Antimicrobial Resistance: A Narrative Review of Published Literature

**DOI:** 10.3390/ijerph17041395

**Published:** 2020-02-21

**Authors:** Tingting Jiang, Xiang-Sheng Chen

**Affiliations:** 1Institute of Dermatology, Chinese Academy of Medical Sciences and Peking Union Medical College, Nanjing 210042, China; jiangtt@ncstdlc.org; 2National Center for STD Control, Chinese Center for Disease Control and Prevention, Nanjing 210042, China

**Keywords:** antimicrobial resistance, outcome impacts, malaria, tuberculosis, HIV, infectious diseases

## Abstract

Antimicrobial resistance (AMR) has become a global threat to not only public health impacts but also clinical and economic outcomes. During the past decades, there have been many studies focusing on surveillance, mechanisms, and diagnostics of AMR in infectious diseases but the impacts on public health, clinical and economic outcomes due to emergence of these AMRs are rarely studied and reported. This review was aimed to summarize the findings from published studies to report the outcome impacts due to AMR of malaria, tuberculosis and HIV and briefly discuss the implications for application to other infectious diseases. PubMed/Medline and Google Scholar databases were used for search of empirical and peer-reviewed papers reporting public health, clinical and economic outcomes due to AMR of malaria, tuberculosis and HIV. Papers published through 1 December 2019 were included in this review. A total of 76 studies were included for this review, including 16, 49 and 11 on public health, clinical and economic outcomes, respectively. The synthesized data indicated that the emergence and spread of AMR of malaria, tuberculosis and HIV have resulted in adverse public health, clinical and economic outcomes. AMR of malaria, tuberculosis and HIV results in significant adverse impacts on public health, clinical and economic outcomes. Evidence from this review suggests the needs to consider the similar studies for other infectious diseases.

## 1. Introduction

Antimicrobial resistance (AMR) is the ability of bacteria, parasites, viruses or fungi to grow and spread in the presence of antimicrobial medicines that are normally active against them [1]. AMR has currently become an alarming threat for human health in the world. A recent report estimated that 10 million deaths will be attributed to AMR by 2050, and 100 trillion USD of the world’s economic outputs will be lost if substantive efforts are not made to contain this threat [2,3]. According to the World Health Organization (WHO) estimates, there were 460,000 people infected with multi-drug resistant tuberculosis (MDR-TB) in 2017 globally [4], and drug resistance continues to complicate the fight against many other infectious diseases, such as HIV, malaria and gonorrhea, as well. As the serious infectious diseases in the world, tuberculosis, HIV and malaria share the common threat of AMR while gonorrhea has been listed as one of the antibiotic-resistant “priority pathogens” by WHO [5]. Resistance to antimalarial drugs has threatened global success in control of malaria since the emergence of resistance to chloroquine in the 1970s [6]. During the 1990s, sulphadoxine-pyrimethamine was introduced in many African countries to replace chloroquine. However, it encountered drug-resistant parasites about a year after this introduction [7]. Artemisinin combination therapies (ACTs) are now recommended as the first-line treatments for uncomplicated *P. falciparum* malaria worldwide. Unfortunately, emergence of artemisinin resistant parasites had been observed recently in Cambodia, Southeast Asia, potentially fostering an increase in malaria cases and deaths [8]. Resistance to streptomycin was detected in a large majority of TB patients treated with this drug as early as in 1940s [9]. Shortly afterwards, a spread of this drug-resistant strains was recognized and continued in an ever wider geographic area despite changing the regime by combining this drug with others. MDR-TB is defined as TB caused by strains of *M. tuberculosis* that are resistant to at least isoniazid and rifampicin [10]. Extensively drug-resistant TB (XDR-TB) is caused by *M. tuberculosis* resistant to at least isoniazid and rifampicin, to any fluoroquinolone and to at least one of three injectable drugs used in anti-TB treatment (capreomycin, kanamycin, amikacin) [11]. HIV strains began to acquire resistance in 1987 when antiretroviral drugs (ARVs) were introduced as therapies for HIV-infected individuals [12]. Since then, a multitude of drug-resistant strains have evolved that differ considerably in their susceptibility to three major classes of ARVs: nucleoside reverse-transcriptase inhibitors (NRTIs), non-nucleoside reverse-transcriptase inhibitors (NNRTIs), and protease inhibitors (PIs). These drug-resistant strains are now being transmitted to individuals who have never received ARVs; that is, transmitted drug resistance has arisen.

Although it is generally believed that AMR could result in significant impacts regarding adverse public health, clinical and economic outcomes due to emergence and spread of AMR, the studies on these issues are limited. This narrative review was aimed to summarize the findings from the published studies on outcome impacts due to AMR of malaria, TB and HIV. 

## 2. Materials and Methods

A literature search was conducted to identify articles published through 1 December 2019 that involved studies on outcome impacts due to AMR of malaria, TB and HIV. The public health outcome was defined as the adverse outcome in terms of increasing transmission and spread of the prolonged infection due to AMR. The clinical outcome was defined as the adverse outcome in terms of increasing treatment failure, mortality and other complications due to AMR. The economic outcome was defined as the adverse outcome in terms of increasing healthcare costs and productivity-loss costs due to AMR. The following terms were used to search articles in PubMed/Medline: (“drug resistance” [MeSH Terms]) AND (malaria [MeSH Terms] OR tuberculosis [MeSH Terms] OR “HIV infections” [MeSH Terms]) AND (“outcome impacts” [All Fields] OR outcome [All Fields] OR impacts [All Fields]). Search of Google Scholar was conducted with the following search strategy: (malaria or tuberculosis or “HIV infections”) AND “drug resistance” AND “outcome”. Titles and abstracts of retrieved records were first screened for inclusion in a full text review. The full texts of potentially relevant studies were then examined to confirm inclusion based on eligibility criteria. We included the articles to meet the following criteria: (a) cross-sectional studies, prospective studies, retrospective studies, case-control studies, meta-analysis, empirical and peer-reviewed studies; (b) at least an abstract with estimates and/or full results published in English; (c) investigate public health, clinical or economic outcomes of AMR of malaria, TB and HIV. The reference list of included articles and relevant systematic reviews were also examined to identify any studies that may have been missed. Commentaries, conference presentations or letter to editors were excluded from this review. The quality of each included article was assesses using the Effective Public Health Practice Project (EPHPP) Quality Assessment Tool for Quantitative Studies [13]. The articles finally included into this review were subsequently categorized according to disease (malaria, tuberculosis and HIV) field of interest (public health, clinical and economic outcomes). Due to wide variation in study designs and populations, we summarized the results of each included article narratively.

## 3. Results

A total of 76 articles were identified as eligible for inclusion in the review (Figure 1). Figure 2 shows the number of articles on each of the three categories and diseases considered in the review. 

The characteristics of the included studies are summarized in Table 1 (16 on public health outcomes), Table 2 (49 on clinical outcomes) and Table 3 (11 on economic outcomes). Studies were predominantly conducted on tuberculosis, accounting for 56.2%, 53.1% and 54.5% on public health, clinical and economic outcomes, respectively, of the totals.

### 3.1. Public Health Outcomes

#### 3.1.1. Malaria

There were two reports that identified drug resistance as the major reason for the occurrence of a malaria epidemic. In Balcad (Somalia), a town with previously low malaria transmission, the incidence of malaria rose more than 20-fold between 1986 and 1988 and the emergence of chloroquine resistance, coupled with favorable meteorological conditions, was identified as the cause [14]. A similar phenomenon occurred in 1994 in Rajasthan, India, and resulted in many deaths [15]. When resistance of *P. falciparum* to sulfadoxine-pyrimethamine was observed in 1990s, an ecological study was conducted in South Africa to assess the association between malaria incidence and the emergence and degree of the resistance to this first-line antimalarial treatment from 1991 to 2001. The data showed that when resistance of P. falciparum to sulfadoxine-pyrimethamine was greater than 10%, the risk ratio for malaria infection was 5.9 compared to the low resistance to sulfadoxine-pyrimethamine period. And the risk of death from malaria was 10.8 and case fatality 1.8 times higher after drug resistance had reached 10%, than before [16]. Since the first two cases of artemisinin resistance were confirmed in Cambodia in 2006, foci of either suspected or confirmed artemisinin resistance have been identified in a few countries. An individual-based malaria transmission model predicted that artemisinin and partner drug resistance at levels similar to those observed in Oddar Meanchey Province in Cambodia could result in an additional 78 million cases over a 5-year period (2016–2020) in Africa, a 7% increase in cases compared to a scenario with no resistance [17].

#### 3.1.2. Tuberculosis

An outbreak of TB in schools of United States caused by *M. tuberculosis* resistant to isoniazid, streptomycin, and para-aminosalicylic acid was reported in 1976. And retrospective investigation revealed that the school outbreak was part of an ongoing community outbreak dating back at least to 1964 [18]. Beginning in 1990, outbreaks of MDR-TB in hospitals in the eastern United States had occurred [19,20,21,22,23]. During 2008–2015, an outbreak of isoniazid-resistant TB with a unique genotype began in Atlanta and spread nationally quickly, provided a warning about the ongoing potential for long-lasting and far-reaching outbreaks, particularly among high-risk populations with untreated HIV co-infection, mental illness, substance abuse, and homelessness [24]. A dynamic state transition model constructed in China showed that by 2050, incidence, prevalence and mortality of Drug-sensitive TB (DS-TB) would decrease by 32%, 50% and 41%, respectively, whereas MDR-TB would increase by respectively 60%, 48% and 35%. Reduction in DS-TB is a result of high treatment and cure rates leading to a decrease in the prevalence of latent tuberculous infection (LTBI), while the increase in MDR-TB is attributed to inappropriate treatment, leading to high transmission of infection and increased LTBI prevalence [25]. Another compartmental model forecasted the percentage of MDR-TB among incident cases of TB to increase, reaching 12.4% in India, 8.9% in the Philippines, 32.5% in Russia, and 5.7% in South Africa in 2040. It also predicted the percentage of XDR-TB among incident MDR-TB to increase, reaching 8.9% in India, 9.0% in the Philippines, 9.0% in Russia, and 8.5% in South Africa in 2040. As prevalence of DR-TB increased, the primary driver of incident DR-TB shifted from acquired drug resistance during treatment to transmission to contacts of individuals with DR-TB, leading to spread of MDR and XDR tuberculosis [26].

#### 3.1.3. HIV

Based on a mathematical model that charted the evolution of drug-resistant HIV from 1996 to 2005 in a San Francisco (CA, USA) cohort of gay men, Blower et al. predicted that drug resistance would reach a high prevalence in developing countries, but that these epidemics of drug-resistant HIV would be driven by resistance acquired within individuals during therapy and not by transmission of resistant strains. Transmitted drug resistance had not increased - and the model predictions revealed that it would not increase - the annual incidence rate in San Francisco. Hence, the results indicated that transmission of resistant strains would be only a minor public health problem in San Francisco [27]. However, in another San Francisco-study presented in 2010 describing an identical population, a theoretical model (the amplification cascade model) revealed that 60% of the resistant strains being transmitted in San Francisco were capable of causing self-sustaining epidemics, and that an individual with an NNRTI-resistant strain can cause, on average, more than one new resistant infection. Consequently, the circulating NNRTI-resistant strains in San Francisco might pose a great and immediate threat to global public health [28]. In sub-Saharan Africa, a Spectrum Goals fast-track model was developed to project the average impact of HIV drug resistance on AIDS deaths and new infections between 2016 and 2030. Results indicated that in a situation where pretreatment drug-resistance levels were generally below 10%, drug resistance was responsible for an estimated 710,000 AIDS deaths and 380,000 new infections by 2030. If levels of pretreatment drug resistance are over 10%, the impact was greater, with an estimated 890 000 AIDS deaths and 450 000 new infections by 2030 attributable to HIV drug resistance [29].

### 3.2. Clinical Outcomes

#### 3.2.1. Malaria

If ineffective first-line treatments were used at health centers and hospital, the time to symptom resolution may be protracted. For example, in Papua New Guinea, the treatment failures for the 28-day follow-up period were 3.8 times more common in 1999 than in 1996, and during the 29–42-day follow-up period 3.8 times more likely in 2002 than in 1991, which were ascribed to the increasing parasite resistance to standard treatment with chloroquine and amodiaquin [30]. As resistance increases, there would be an increase in the number of patients who present with severe malaria, thus case fatality rate would rise. Retrospective and prospective hospital-based studies in various African countries had documented significant increase (from 2- to 11-fold) in admissions for severe malaria and malaria deaths when chloroquine resistance developed and spread [31,32,33,34,35,36,37,38]. The malaria-related effects on pregnant women, their fetuses, and newborns also comprise an extremely large and often hidden burden [89,90]. In a study among pregnant women and newborn infants in Timika, it was found that in those with history of malaria during pregnancy, the increasing use of dihydroartemisinin-piperaquine, the first-line malaria treatment instead of chloroquine [91], was associated with a 54% fall in the proportion of maternal malaria at delivery and a 98% decrease in congenital malaria. Also, the change to more effective treatment was associated with an absolute 2% reduction of maternal severe anaemia and absolute 4.5% decrease in low birth weight babies [39]. Three prospective cohort studies found that patients from western Cambodia, where has been the focus of artemisinin-resistant malaria, were more likely to had parasite recrudescence [8,40,41]. In Silico Model for Antimalarial Drug Treatment and Failure showed that the development of artemisinin tolerance and resistance will have an immediate, large impact on the likely clinical efficacy of artemisinin combination therapies [42].

#### 3.2.2. Tuberculosis

According to the WHO guidelines, there are six treatment outcomes in patients with TB: cured, treatment completed, died, treatment failed, defaulted, and transferred [43]. Two studies from Saudi Arabia found that the prevalence of rifampicin resistance was significantly higher in those with failed treatment than in successfully treated patients [44,45]. A retrospective study reported a relative risk of death of 1.9 among patients with rifampicin resistance compared with those with pan-susceptible group [46]. Five studies reported an increase (from 1.4- to 12.4-fold) of poor treatment outcomes (death, failure or default from TB therapy) in isoniazid mono-resistance patients compared to isoniazid sensitive case [45,46,47,48,49,50]. Patients with MDR-TB had considerably poorer clinical outcomes than if they developed disease due to drug-susceptible strains. Compared with susceptible cases, a significant increase (from 3- to 395-fold) in unfavorable outcomes was recorded among MDR-TB cases [43,44,45,51,52,53,54,55,56,57,58,59,60,61,62,63,64]. And the survival time in patients with MDR-TB was significantly shorter compared to those without MDR-TB [65,66]. Using country as the unit of analysis, an ecologic study showed that failure rates averaged 5.0%, and recurrence rates averaged 12.8% in the 20 countries where prevalence of initial MDR exceeded 3%, compared with an average of 1.6% and 8.1%, respectively, in 83 countries where initial MDR prevalence was less than 3% [67]. In a meta-analysis, it showed a stepwise worsening of treatment outcomes in MDR-TB cases treated in multiple centers as the resistance pattern of infecting TB strains advanced from MDR without additional resistance, to added resistance to a second-line injectable drug, to a fluoroquinolone, and then to both (XDR-TB). The study reported that compared with poor treatment outcomes, treatment success was higher in MDR-TB patients infected with strains without additional resistance (64%) or with resistance to second-line injectable drugs only (56%), than in those having resistance to fluoroquinolones alone (48%) or to fluoroquinolones plus second-line injectable drugs (40%) [68].

#### 3.2.3. HIV

When using antiretroviral drugs as therapies for HIV-infected individuals, the antiretroviral treatment was considered efficient if the viral load decreases or remained undetectable (<50 copies/mL). The emergence of drug resistance, however, was reported to influence the virological response. In a study from Canada, it found that baseline drug resistance phenotype was predictive of poor virological response to dual protease inhibitor combination. Patients were at least four times less likely to achieve a decrease in plasma HIV RNA viral load if their viral isolates were resistant to ritonavir or saquinavir [69]. Cohort studies from USA reported significant increase (from 1.5- to 12.4-fold) of virological failure in subjects with baseline NNRTI resistance compared to subjects without resistance [70,71,72]. Studies from Europe also reported a significant association between NRTI resistance and therapy response [73,74]. Using the Stanford HIVdb algorithm, which assigns genotypic sensitivity score (GSS) of 1.00, 0.75, 0.50, 0.25 and 0.00 to the five levels of resistance (susceptible, potential low-level, low-level, intermediate-level and high-level resistance, respectively), a study from Taiwan found that compared with regimens with GSS >2.5, initiation of regimens with GSS ≤2.5 was associated with a higher treatment failure rate (39.3% versus 15.7%) and shorter time to treatment failure [75]. Experiencing virological failure with drug resistance is a prognostic sign for poorer long-term clinical outcome. Studies from multiple countries have documented the association between drug resistance and an increased risk of death or new AIDS defining event/death. The 48-month proportion of patients with none, one, two or three class-wide resistance were 8.9, 11.7, 13.4 and 27.1%, respectively, for death; 6.1, 9.9, 13.4 and 21.5%, respectively, for AIDS-related death; and 16.0, 17.7, 19.3 and 35.9%, respectively, for new AIDS event/death [76]. By 96 months from baseline, the proportion of patients with a new AIDS diagnosis or death was 20.3% in patients with no evidence of virological failure and 53% in those with virological failure and mutations to three drug classes [77]. Detection of drug resistance, particularly if extended to all three drug classes was related to poorer clinical outcome and represented a risk-marker of disease progression and death.

### 3.3. Economic Outcomes

#### 3.3.1. Malaria

In areas with high rates of chloroquine-resistance, physicians and governments have changed empiric therapy for malaria, increasing overall treatment costs. Phillips and Phillips-Howard calculated that the use of quinine versus chloroquine as first-line therapy in 150 million patients with malaria would increase spending by as much as $100 million [78]. When physicians use ineffective drugs as empiric therapy, resistance will manifest itself clinically as treatment failure, and costs, morbidity, and mortality will be greater for patients with infections due to resistant organisms than for patients with infections due to susceptible organisms. Based on a decision tree model applied to all malaria-endemic countries using their specific estimates for malaria incidence, transmission intensity and GDP, Lubell et al. projected an excess of 116,000 deaths annually in the scenario of widespread artemisinin resistance. And the predicted medical costs for retreatment of clinical failures and for management of severe malaria exceed $32 million per year. Productivity losses resulting from excess morbidity and mortality were estimated at $385 million for each year during which failing ACT remained in use as first-line treatment [79].

#### 3.3.2. Tuberculosis

A South African cost analysis showed that the per-patient cost of XDR-TB was $26,392, four times greater than MDR-TB ($6772), and 103 times greater than DS-TB ($257). Despite DR-TB comprising only 2.2% of the case burden, it consumed ~32% of the total estimated 2011 national TB budget of US $218 million [80]. In Asia, a cross-sectional survey in Cambodia showed that the mean total household cost for TB patients was $477, compared to $1525 in MDR-TB cases [81]. In Europe, a retrospective study from London found that the mean cost of managing a case of pulmonary MDR-TB was in excess of £60,000, 10 times the cost of treating drug-sensitive disease (£6040) [82]. In South America, a study from Ecuador found that total non-MDR-TB related patient costs averaged $960 per patient, compared to an average total cost of $6880 for MDR-TB patients, which represented respectively 31% and 223% of the average Ecuadorian annual income [83]. In North America, a US study States showed that average hospitalization cost per XDR-TB patient ($285,000) was 3.5 times that per MDR-TB patient ($81,000). Hospitalization episode costs for MDR-TB ranked third highest and those for XDR-TB highest among 262 principal diagnosis categories [84]. Another retrospective study reported that total direct costs averaged $430,000 per XDR-TB patient, three times greater than MDR-TB ($13,000), and 25 times greater than non-MDR TB ($17,000). Direct-plus-productivity-loss costs averaged $554,000 per XDR-TB patient, two times greater than MDR-TB patient ($260,000). Applying these averages to 364 cases of MDR-TB and nine cases of XDR-TB in the US during 2005–2007, direct costs were ≈ $53 million and direct-plus-productivity-loss costs were ≈ $100 million [85].

#### 3.3.3. HIV

Only one study reported healthcare costs associated with antiretroviral drug resistance. The study from Canada reported that patients with no resistance had mean per patient per month (PPPM) costs of Canadian dollars (CDN) 1058, by contrast with the CDN 1291 costs of patients with secondary/acquired resistance. And the mean costs increased by number of resistant classes: mean costs for one, two or three ARV class resistance was CDN 1278, 1337 and 1801, respectively [86]. Considering that treatment failure and switching may follow from the development of drug resistance, studies of the costs due to treatment failure or treatment changes could potentially be extrapolated to represent the costs of resistance [92]. A report demonstrated total annual costs ranged from $31,700 for a patient on a 1st-line highly active antiretroviral therapy (HAART) regimen to $42,600 on 6th-line [87]. In a separate study presented in 2010, over a follow-up period of up to 60 months, total mean healthcare costs were $35,000 higher for patients on a 3rd- or greater line treatment regimen compared to patients on a 1st- or 2nd-line treatment regimen [88].

### 3.4. Summary of Evidence

By taking into account the number and evidence quality of the included studies, the synthesized findings from these studies show that there is strong evidence that drug-resistant tuberculosis has resulted in adverse public health, clinical and economic outcomes, drug-resistant malaria has resulted in adverse public health and clinical outcomes, and drug-resistant HIV infection has resulted in adverse clinical outcomes (Table 4). 

Further higher quality research on the public health outcome of drug-resistant HIV infection and economic outcome of drug-resistant malaria are needed.

## 4. Discussion

This review exposes the multi-dimensional outcomes due to emergence and spread of AMR of malaria, tuberculosis and HIV to the antimicrobials recommended and/or used for treatment of certain infections. Owing to the increased burden of these infectious diseases in developing countries, most of the studies were conducted in these countries, particularly on clinical outcomes due to AMR. of the three diseases included in this review, tuberculosis’s outcomes due to AMR are mostly recognized by researchers to conduct studies accordingly, which is probably because tuberculosis has a relatively high mortality—the leading cause of death from a single infectious agent (above HIV/AIDS) and AMR rate [4].

It is well noted that the emergence of AMR or spread of resistant strain might result in a failure or delay to eliminate the infection, affecting the likelihood of onward transmission [93]. Regarding public health or epidemiological outcomes due to AMR, evidences were mostly obtained from population-based cohort studies or cross-sectional studies before 2000. However, mathematical models were increasingly used to predict the public health burden due to AMR in recent years. Background data and assumption parameters are critical for development of the models and outcomes predicted from the models. For example, in a San Francisco cohort of gay men discrepancies regarding the epidemics of drug-resistant HIV were found from model predictions. These differences in levels of resistance and effects on HIV incidence might be due to different time horizons, different model assumptions and other extraordinary factors or new events, such as vaccines, diagnostic tools, and policy changes [25]. Built upon current disease estimates and trend assumptions, considering the source, course and transmission of infection, and taking other major forces into account, models have the potential to be particularly valuable in predicting the public health outcomes of drug resistance and the future disease burden. The relationship between the emergence of drug resistance in patients and the risk of clinical outcomes has been subject to various analyses from cohort studies. Antimicrobial-resistant organisms affect patient outcomes in several ways. Treatment factors including delay in appropriate antimicrobial treatment, decreased antimicrobial effectiveness, increased antimicrobial toxicity, improper antimicrobial dosing and/or increased need for surgery and other invasive procedures may contribute to adverse outcomes in patients infected with these organisms [94]. Despite of the use of different methods of analysis as well as other factors, the results of studies included in this review have shown that patients with infections caused by drug-resistant organism generally had an increased risk of worse clinical outcomes and death, than patients infected with the same organism not demonstrating the resistance pattern. However, many studies were conducted in single institution and had small sample size. Future studies estimating clinical outcomes of antimicrobial-resistant infections should address the methodological limitations by using large patient populations in multicentre settings or by using large administrative datasets. There was a wide range of cost estimates across the studies probably because of the differences in economic background in study population and areas, methodologies used to collect the data and estimate the costs, comparator (infection susceptible to antimicrobials, or no infection) to be used for estimating additional cost, and study settings. Despite of these, additional expenditure resulted from emergence and spread of AMR was consistently reported in many of these studies. The economic burden of antimicrobial-resistant organisms can be assessed from a number of different perspectives, including that of society, the hospital, a third-paid party, a government agency or the patient [94]. Most studies in this review did not attempt to measure the cost of resistance from a societal perspective (including all the economic impacts on patients, physicians, health care providers, third-party buyers, drug manufacturers, and overall societal welfare), making the true economic burden of antimicrobial-resistant infections difficult to quantify accurately. To better quantify the economic repercussions of AMR, future studies should consider costs from multiple perspectives [95].

Findings from the current review have important implications for planning studies to address the public health, clinical and economic outcome impacts in other infectious diseases including gonorrhea. *Neisseria gonorrhoeae*—a pathogen causing gonorrhea has shown the capacity to develop resistance to the antimicrobials introduced for treatment of this infection since sulphonamides was firstly introduced for treatment of gonorrhea in the 1930s. Currently, *N. gonorrhoeae* is resistant to most antimicrobials previously recommended for treatment and emergence of resistance to the third-generation extended-spectrum cephalosporins has been found in many settings [96]. Gonorrhoea, together with malaria, TB and HIV have been listed by WHO as the global priority concerns to address their AMR [5,96]. No doubt efforts to improve surveillance programme for monitoring trends of gonococcal resistance, explore resistant mechanisms for responding to emergence of the resistance, and develop new antimicrobials for providing alternative treatment options are important. However, studies on outcome impacts due to gonococcal resistance are most needed to justify the investment to address the AMR in gonorrhea. However, evidence from studies on outcome impacts of gonococcal resistance [97,98] is relatively limited. Actually, investment to control of gonorrhea is disproportionately lower than many of other infectious diseases [99]. The findings from this review on malaria, tuberculosis and HIV may be helpful for considering the similar studies in gonorrhea. This consideration has been firstly highlighted in China and included into its comprehensive ROADMAP plan to address research needs for gonococcal AMR [100].

There are several limitations to this narrative review. First, we only included two databases in our search. Second, our search strategy did not extend to grey literature, and were unable to rule out publication bias. Third, due to wide variations in study design, publication years and populations, it is hard for us to make a pooled estimate of AMR outcome for each disease and consequently to do comparisons in terms of AMR outcome impacts among these three diseases, which also points towards the need for more harmonised methods to assess public health, clinical and economic outcome impacts due to AMR among different infectious diseases.

## 5. Conclusions

In conclusion, previous studies indicate that emergence and spread of AMR of malaria, TB and HIV have resulted in adverse public health, clinical and economic outcomes. Limited evidence from this review suggest the needs to consider the similar studies for other infectious diseases including gonorrhea.

## Figures and Tables

**Figure 1 ijerph-17-01395-f001:**
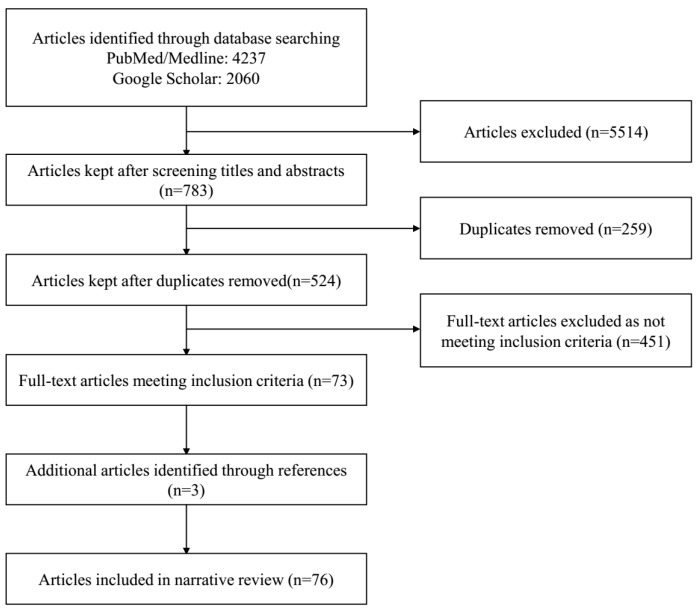
PRISMA flow diagram.

**Figure 2 ijerph-17-01395-f002:**
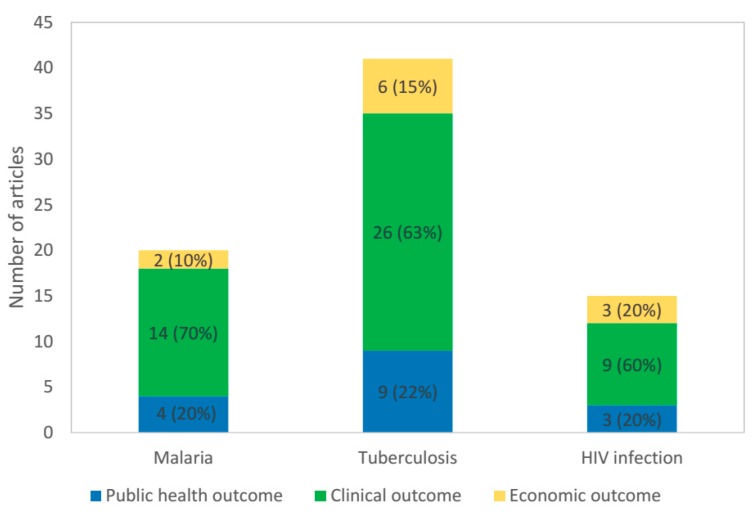
Articles finally included into narrative review were categorized according to disease (malaria, tuberculosis and HIV) field of interest (public health, clinical and economic outcomes).

**Table 1 ijerph-17-01395-t001:** Characteristics of included studies on public health outcomes due to AMR.

Author by Disease	Study Year	Study Site	Sample Size	Study Design/Method	Outcome	Quality Assessment
Malaria						
Warsame et al. [14]	1982–1988	Somalia	109	Retrospective study	The emergence of chloroquine resistance, accelerated by high drug pressure, low herd immunity and favourable meteorological conditions were identified as major causes of the epidemic of falciparum malaria in Balcad, Somalia.	STRONG
Sharma et al. [15]	1994	India	32	Cross-sectional study	In 1994, there was a falciparum malaria epidemic in Rajasthan, India, with many deaths, and most of the parasite isolates (95%) were resistant to chloroquine.	STRONG
Knight et al. [16]	1982–1988, 1991–2001	South Africa	600000	Ecological study	The relative risk for malaria infection after the level of drug resistance reached 10% was 4.5 (95% CI: 4.0–5.2) in the chloroquine period and 5.9 (95% CI: 5.7–6.1) in the sulfadoxine-pyrimethamine period.	MODERATE
Slater et al. [17]	2016–2020	Africa	-	An individual-based malaria transmission model	Artemisinin and partner drug resistance at levels similar to those observed in Oddar Meanchey province in Cambodia could result in an additional 78 million cases over a 5 year period, a 7% increase in cases compared to a scenario with no resistance.	WEAK
Tuberculosis						
Reves et al. [18]	1976	USA	15	Retrospective investigation	An outbreak of tuberculosis in 1976 was caused by mycobacteria resistant to isoniazid (INH), streptomycin (SM), and para-aminosalicylic acid (PAS).	STRONG
Ussery et al. [19]	1995	USA	18	Cohort study	A trend between TST conversion and participation in autopsies on persons with MDR-TB was observed.	STRONG
Coronado et al. [20]	1990–1991	USA	16	Cross-sectional study	From January 1990 to December 1991, 16 patients with multidrug-resistant tuberculosis (MDR-TB) were caused by nosocomial transmission of MDR-TB resistant to isoniazid, rifampin, and streptomycin.	STRONG
Friedman et al. [21]	1995	USA	167	Cross-sectional study	Forty-three (34%) of 127 drug-susceptible isolates and 19 (79%) of 24 multidrug-resistant isolates had RFLP patterns representing possible recent exogenous infection.	STRONG
US CDC. [22]	1990–1991	USA	NR	Retrospective study	During 1990 and 1991, outbreaks of multidrug-resistant tuberculosis (MDR-TB) in four hospitals (one in Miami and three in New York City) were due to nosocomial transmission among HIV-infected persons.	STRONG
Nivin et al. [23]	1998	USA	24	Cross-sectional study	Nosocomial transmission appeared to account for an increase in cases of multidrug-resistant tuberculosis (MDRTB) at a large urban facility where a prior nosocomial outbreak of MDRTB had occurred.	STRONG
Powell et al. [24]	2008–2015	USA	110	Retrospective investigation	Of 110 outbreak cases in Georgia, 86 (78%) were culture confirmed and isoniazid resistant.	STRONG
Mehra et al. [25]	2010–2050	China	-	Dynamic state transition model	In China, by 2050, incidence, prevalence and mortality of MDR-TB will increase by respectively 60%, 48% and 35%, which attributed to inappropriate treatment, leading to high transmission of infection and increased LTBI prevalence.	WEAK
Sharma et al. [26]	2000–2040	India, Philippines, Russia and South Africa	-	A compartmental model	The model forecasted the percentage of MDR tuberculosis among incident cases of tuberculosis to increase, reaching 12.4% (95% prediction interval 9.4–16.2) in India, 8·9% (4.5–11.7) in the Philippines, 32.5% (27.0–35.8) in Russia, and 5·7% (3.0–7.6) in South Africa in 2040. It also predicted the percentage of XDR tuberculosis among incident MDR tuberculosis to increase, reaching 8·9% (95% prediction interval 5.1–12.9) in India, 9·0% (4.0–14.7) in the Philippines, 9·0% (4.8–14.2) in Russia, and 8·5% (2.5–14.7) in South Africa in 2040.	WEAK
HIV						
Blower et al. [27]	1996–2005	USA	-	A mathematical model	The epidemic of resistance is being generated mainly by the conversion of drug-sensitive cases to drug-resistant cases, and not by the transmission of resistant strains.	WEAK
Smith et al. [28]	2008–2013	USA	-	A biologically complex multistrain network model	60% of the currently circulating ARV-resistant strains in San Francisco are capable of causing self-sustaining epidemics, because each individual infected with one of these strains can cause, on average, more than one new resistant infection.	WEAK
Phillips et al. [29]	2016–2030	Sub-Saharan Africa	-	HIV Synthesis Model	In a situation in which current levels of pretreatment HIVDR are over 10% (mean, 15%), 16% of AIDS deaths (890 000 deaths), 9% of new infections (450 000), and 8% ($6.5 billion) of ART program costs in SSA in 2016–2030 will be attributable to HIVDR.	WEAK

NR, not reported.

**Table 2 ijerph-17-01395-t002:** Characteristics of included studies on clinical outcomes due to AMR.

Author by Disease	Study Year	Study Site	Sample Size	Study Design/Method	Outcome	Quality Assessment
Malaria						
Nsanzabana et al. [30]	1991–2002	Papua New Guinea	6678	Retrospective study	Treatment failure rates multiplied by 3.5 between 1996 and 2000 but then decreased dramatically after treatment policy change.	STRONG
Khoromana et al. [31]	1978–1983	Malawi	224	Retrospective study	Parasitological failure ranged from 41–65% following administration of chloroquine 25 mg (base)/kg.	MODERATE
Greenberg et al. [32]	1982–1986	Zaire	6208	Retrospective study	The proportional malaria admission rate increased from 29.5% in 1983 to 56.4% in 1986, and the proportional malaria mortality rate, from 4.8% in 1982 to 15.3% in 1986, which were temporally related to the emergence of chloroquine-resistant Plasmodium falciparum malaria in Kinshasa.	STRONG
Carme et al. [33]	1983–1989	Congo	NR	A retrospective and prospective hospital-based study	The results show a marked increase in hospitalizations for malaria, noticeable since 1985, and which now account for about 50% of the overall non-surgical hospitalizations.	STRONG
Asindi et al. [34]	1986–1988	Nigeria	134	Retrospective study	Malaria was the dominant cause (73%) of febrile convulsion (FC); 81% of these cases did not respond to chloroquine.	STRONG
Zucker et al. [35]	1991	Kenya	1223	Prospective study	Treatment for malaria with chloroquine was associated with a 33% case fatality rate compared with 11% for children treated with more effective regimens (pyrimethamine/sulfa, quinine, or trimethoprim/sulfamethoxazole for five days).	STRONG
Shanks et al. [36]	1980–1997	Kenya	10169	Retrospective study	The dramatic increases in the numbers of malaria admissions (6.5 to 32.5% of all admissions), case fatality (1.3 to 6%) and patients originating from low-risk, highland areas (34 to 59%) were probably due to chloroquine resistance during the late 1980s in the subregion.	STRONG
Zucker et al. [37]	1991–1994	western Kenya	1223	Prospective study	The trend in case-fatality rates for malaria decreased as an increasing proportion of children received an effective treatment regimen; adjusted malaria case-fatality rates were 5.1%, 3.6%, and 3.3% in 1992, 1993, and 1994, respectively, when 85% of children in 1992 and 97% of children in 1993–1994 received effective therapy.	STRONG
Brewster et al. [38]	1988–1990	Gambia	9584	Prospective study	With the emergence of chloroquine-resistant malaria over the 3 years, there was a 27% annual increase in severe anaemia owing to malaria.	STRONG
Poespoprodjo et al. [39]	2004–2010	Indonesia	7744	Prospective cohort study	In those with history of malaria during pregnancy, the increasing use of DHP was associated with a 54% fall in the proportion of maternal malaria at delivery and a 98% decrease in congenital malaria (from 7.1% prior to 0.1% after policy change).	STRONG
Amaratunga et al. [8]	2012–2013	Cambodia	241	Prospective cohort study	In Pursat, where artemisinin resistance is entrenched, 37 (46%) of 81 patients had parasite recrudescence. In Preah Vihear, where artemisinin resistance is emerging, ten (16%) of 63 patients had recrudescence and in Ratanakiri, where artemisinin resistance is rare, one (2%) of 60 patients did.	STRONG
Leang et al. [40]	2011–2013	Cambodia	425	A prospective multicenter open-label study	The most significant risk factor associated with DHA-PP treatment failure was infection by parasites carrying the K13 mutant allele (odds ratio [OR], 17.5; 95% confidence interval [CI], 1 to 308; *p* = 0.04).	STRONG
Leang et al. [41]	2008–2011	Cambodia	438	Prospective cohort study	In 2010, the PCR-corrected treatment failure rates for DP on day 42 were 25% (95% confidence interval [CI] = 10 to 51%) in Pailin and 10.7% (95% CI = 4 to 23%) in Pursat, while the therapeutic efficacy of DP remained high (100%) in Ratanakiri and Preah Vihear provinces, located in northern and eastern Cambodia.	STRONG
Winter et al. [42]	NR	NR	-	In Silico Model for Antimalarial Drug Treatment and Failure	The development of artemisinin tolerance and resistance will, unless checked, have an immediate, large impact on the protection afforded to its partner drug and on the likely clinical efficacy of artemisinin combination therapies.	WEAK
Tuberculosis						
Espinal et al. [43]	1994–1996	Dominican Republic, Hong Kong Special Administrative Region, Italy, Ivanovo Oblast, the Republic of Korea, and Peru	6402	Retrospective cohort study	Treatment failure (relative risk [RR], 15.4; 95% confidence interval [CI], 10.6–22.4; *p* < 0.001) and mortality (RR, 3.73; 95% CI, 2.13–6.53; *p* < 0.001) were higher among new multidrug-resistant TB cases than among new susceptible cases.	STRONG
Singla et al. [44]	1998–1999	Saudi Arabia	515	Retrospective cohort study	Sputum smear conversion rates at the end of 3 months of treatment in patients with any rifampicin resistance or with multidrug resistance were inferior to those of patients with sensitive strains (89.8% vs. 96.3%, *p* = 0.016 and 80% vs. 96.3%, *p* = 0.008, respectively).	MODERATE
Samman et al. [45]	1993–1999	Saudi Arabia	147	Retrospective cohort study	The prevalence of poor compliance and multiply drug-resistant Mycobacterium tuberculosis were found to be significantly higher among those with treatment failure than among those in whom treatment was successful.	STRONG
Anuwatnonthakate et al. [46]	2004–2008	Thailand	9736	Retrospective cohort study	Cox regression analysis showed a significantly higher risk of death among patients with rifampicin resistance (adjusted hazard ratio (aHR) 1.9, 95% confident interval (CI), 1.5–2.5) and isoniazid monoresistance (aHR 1.4, 95% CI 1.1–1.7) than those with pan-susceptible group.	STRONG
Deepa et al. [47]	2011	India	1947	Retrospective record review	Of 144 INH resistant cases, 64 (44%) had poor treatment outcomes (25 (17%) default, 22 (15%) death, 12 (8%) failure and 5 (3%) transfer out) as compared to 287 (31%) among INH sensitive cases [aRR 1.46; 95% CI (1.19–1.78)].	STRONG
Báez-Saldaña et al. [48]	1995–2010	Southern Mexico	1243	Prospective cohort study	IMR patients had a higher probability of failure (adjusted hazard ratio (HR) 12.35, 95% CI 3.38–45.15) and death due to TB among HIV negative patients (aHR 3.30. 95% CI 1.00–10.84).	STRONG
Nagu et al. [49]	2010–2011	Tanzania	1365	A multicentre, prospective observational study	Isoniazid resistance [relative risk (RR) = 6.0; 95% CI = 1.9–18.7; *p* < 0.01] was an independent predictor of poor treatment outcomes.	STRONG
Karo B et al. [50]	2002–2014	European Union/European Economic Area	194948	Retrospective cohort study	Treatment success was lower among INH mono-resistant cases (Odds ratio (OR): 0.7; 95% confidence interval (CI): 0.6–0.9; adjusted absolute difference in treatment success: 5.3%).	STRONG
García-García et al. [51]	1995–1998	Southern Mexico	2525	Prospective cohort study	Patients with multi–drug-resistant TB had a significantly poorer prognosis than patients with fully susceptible strains or with other resistant strains (*p* = 0.03).	STRONG
García-García et al. [52]	1995–1999	Mexico	387	Prospective cohort study	Cox-adjusted relative risks showed that MDR (RR 2.5, 95%CI 1.02–6.16, *p* = 0.04)was associated with mortality, controlling for age.	STRONG
García-García et al. [53]	1995–1999	Southern Mexico	371	Prospective cohort study	Patients with drug resistance had a higher probability of treatment failure (OR = 16.9, CI 95% 4.5–63.0) and patients with MDR strains had a higher probability of need of re-treatment (RR = 24.4, CI 95% 8.8–67.6), and of death (RR = 4.0, CI 95% 1.5–10.7).	STRONG
Noeske et al. [54]	1997–1998	Cameroon	560	Retrospective cohort study	332 of the 410 patients (81%) with DS-TB were cured, compared to 109/150 (72.7%) patients with DR-TB (odds ratio [OR] = 0.62, 95% confidence interval [CI] 0.40–0.99). Seven patients (1.7%) failed treatment in the DS-TB group vs. 9 (6.0%) in the DR-TB group (OR = 3.67, 95% CI 1.23–11.18). No significant difference was found in rates of death, default or transfer.	STRONG
Toungoussova et al. [55]	1999	Russia	235	Retrospective cohort study	The high rates of death (16.7%) and failure (66.7%) among patients infected with multidrug-resistant strains illustrate the negative impact of multidrug resistance on the outcome of tuberculosis treatment. Pan-resistance was significantly associated with treatment failure (*p* < 0.001).	STRONG
Ohkado et al. [56]	2000	Philippines	457	Cross-sectional survey & cohort analysis of treatment	Over 90% of the new cases, either pan-susceptible or mono-resistant, were successfully treated with the standard regimen, but four of nine MDR new cases could not be cured.	STRONG
Cox et al. [57]	2001–2002	Uzbekistan	213	Retrospective observational study	Mortality was high, with an average of 15% (95% confidence interval, 11% to 19%) dying per year after diagnosis (6% of 73 pansusceptible cases and 43% of 55 MDR TB cases also died per year).	STRONG
Matos et al. [58]	2001–2003	Brazil	396	Prospective cohort study	An association was found between resistance and mortality from tuberculosis (adjusted OR: 7.13; 95%CI: 2.25–22.57; *p* < 0.001).	STRONG
Seddon et al. [59]	2003–2009	South Africa	142	Prospective cohort study	Multidrug-resistant tuberculosis (adjusted odds ratio: 12.4 [95% confidence interval: 1.17–132.3]; *p* = 0.037) was a risk factor for unfavorable outcome, and multidrug-resistant tuberculosis remained a risk for death (adjusted odds ratio: 63.9 [95% confidence interval: 4.84–843.2]; *p* = 0.002).	STRONG
Sun et al. [60]	2010	China	234	Cohort study	Nine years after the diagnosis of TB, 69 or 29.5% of the 234 patients had died (32 or 21.6% of non-MDR-TB versus 37 or 43.0% of MDR-TB) and the overall mortality rate was 39/1000 per year (PY) (27/1000 PY among non-MDR versus 63/1000 PY among MDR-TB).	STRONG
Sun et al. [61]	2010	China	250	Cohort study	The mean time for recurrence among MDR-TB patients was 5.7 years, compared to 7.2 years among non-MDR-TB patients.	STRONG
Lockman et al. [62]	1998	Estonia	103	Retrospective observational study	MDR tuberculosis (hazard ratio [HR], 7.8; 95% CI, 1.6–37.4) was associated with death due to tuberculosis in multivariable analysis.	STRONG
Quy et al. [63]	1998–2000	Vietnam	2293	Retrospective cohort study	Failure was associated with multidrug resistance (adjusted odds ratios [aOR] 49.6 and 16.6, respectively) and combined resistance to isoniazid (INH) and streptomycin (SM) (aOR 13.4 and 4.8)	STRONG
Pradipta et al. [64]	2005–2015	Netherlands	10303	Retrospective cohort study	Among all DR-TB cases, patients with Multi Drug-Resistant Tuberculosis (MDR-TB) (OR 4.43; 95% CI 1.70–11.60) were more likely to have unsuccessful treatment.	STRONG
Eyob et al. [65]	1999–2001	Ethiopia	490	Prospective cohort study	Among HIV-infected TB patients who died during follow-up, survival time in those with a resistant Mycobacterium tuberculosis strain was significantly shorter compared to those with a sensitive strain (6 vs. 13 months).	STRONG
Sungkanuparph et al. [66]	1999–2004	Thailand	225	Retrospective cohort study	INH resistance, RMP resistance and MDR-TB were associated with shorter survival (log-rank test, *p* < 0.005). MDR-TB (hazard ratio [HR] 11.7; 95% CI 2.1–64.9) was significant risk factors for death.	STRONG
Mak et al. [67]	2003 & 2004	155 countries	121 countries	Ecologic study	Among countries using one of two standardized initial regimens, failure rates averaged 5.0%, and relapse rates averaged 12.8% in the 20 countries where prevalence of initial multidrug resistance exceeded 3%, compared with an average of 1.6% (*p* < 0.0001) and 8.1% (*p* = 0.0002), respectively, in 83 countries where initial multidrug resistance prevalence was less than 3%.	WEAK
Falzon et al. [68]	1980–2009	31 centres	-	Meta-analysis	Compared with treatment failure, relapse and death, treatment success was higher in MDR-TB patients infected with strains without additional resistance (*n* = 4763; 64%, 95% CI 57–72%) or with resistance to second-line injectable drugs only (*n* = 1130; 56%, 95% CI 45–66%), than in those having resistance to fluoroquinolones alone (*n* = 426; 48%, 95% CI 36–60%) or to fluoroquinolones plus second-line injectable drugs (extensively drug resistant (XDR)-TB) (*n* = 405; 40%, 95% CI 27–53%).	MODERATE
HIV						
Harrigan et al. [69]	1996–1997	Canada	297	prospective cohort study	Patients classified as resistant to either drug using either method had median decreases in plasma viral load of 0.05 log10 HIV RNA copies/mL or less, compared to >0.8 log10 for those with sensitive virus.	STRONG
Kuritzkes et al. [70]	2008	USA	220	A case-cohort study	The risk of virologic failure for subjects with baseline NNRTI resistance was higher than that for subjects without such resistance (hazard ratio 2.27 [95% confidence interval], 1.15–4.49; *p* = 0.018).	STRONG
Taniguchi et al. [71]	2001–2009	USA	801	Retrospective study	In multivariate analysis, nonnucleoside reverse transcriptase inhibitor (NNRTI) resistance was associated with a 1.5-fold increased risk of virologic failure.	STRONG
Simen et al. [72]	2005	USA	491	Cohort study	The risk of VF was higher for those who had an NNRTI-resistance mutation detected by both methods (hazard ratio [HR], 12.40 [95% confidence interval {CI}, 3.41–45.10]) and those who had mutation(s) detected only with ultra-deep sequencing (HR, 2.50 [95% CI, 1.17–5.36]).	STRONG
Miller et al. [73]	1998	Germany	43	Cross-sectional study	After adjustment for all variables, phenotypic resistance to zidovudine remained the only significantly associated factor of therapy response.	STRONG
Derdelinckx et al. [74]	2000	Belgium	93	Retrospective study	In a multivariate logistic model, controlled for log VL and CD4 count at treatment start, the association of transmitted resistance with treatment failure remained significant (OR: 148, 95% CI: 3.34- > 999.9, *p* = 0.027).	STRONG
Lai et al. [75]	2000–2010	Taiwan	1349	Matched case-control study	Compared with regimens with GSS >2.5, initiation of regimens with GSS ≤2.5 was associated with a higher treatment failure rate (39.3% versus 15.7%, *p* = 0.02) and shorter time to treatment failure (log-rank *p* < 0.001).	STRONG
Zaccarelli et al. [76]	1999–2003	Italy	623	Observational, longitudinal cohort study	Kaplan-Meier analyses for end-points at 48 months in patients with no CWR, one CWR, two CWR or three CWR were 8.9, 11.7, 13.4 and 27.1%, respectively, for death; 6.1, 9.9, 13.4 and 21.5%, respectively, for AIDS-related death; and 16.0, 17.7, 19.3 and 35.9%, respectively, for new AIDS event/death.	STRONG
Cozzi-Lepri et al. [77]	2003–2007	UK	8229	Cohort study	By 96 months from baseline, the proportion of patients with a new AIDS diagnosis or death was 20.3% (95% CI:17.7–22.9) in patients with no evidence of virological failure and 53% (39.3–66.7) in those with virological failure and mutations to three drug classes (*p* = 0.0001)	STRONG

NR, not reported.

**Table 3 ijerph-17-01395-t003:** Characteristics of included studies on economic outcomes due to AMR.

Author by Disease	Study Year	Study Site	Sample Size	Study Design/Method	Outcome	Quality Assessment
Malaria						
Phillips et al. [78]	1996	NR	NR	Simplifying assumptions and illustrative calculations	The use of quinine versus chloroquine as first-line therapy in 150 million patients with malaria would increase spending by as much as $100 million.	WEAK
Lubell et al. [79]	2014	All malaria-endemic areas	-	Decision tree model	The predicted medical costs for retreatment of clinical failures and for management of severe malaria exceed US$32 million per year. Productivity losses resulting from excess morbidity and mortality were estimated at US$385 million for each year during which failing ACT remained in use as first-line treatment.	WEAK
Tuberculosis						
Pooran et al. [80]	2011	South Africa	-	Cost analysis	Assuming adherence to national DR-TB management guidelines, the per patient cost of XDR-TB was $26,392, four times greater than MDR-TB ($6772), and 103 times greater than drug-sensitive TB ($257).	WEAK
Pichenda et al. [81]	2011	Cambodia	277	Cross-sectional survey	The mean total household cost for TB patients was $477, compared to $1525 in MDR-TB cases.	STRONG
White et al. [82]	1996–1999	UK	9	Retrospective study	The mean cost of managing a case of pulmonary MDR TB was in excess of 60,000 pounds sterling and for sensitive disease it was 6040 pounds sterling.	STRONG
Rouzier et al. [83]	2007	Ecuador	118	Cross-sectional survey	Among 104 non-MDR-TB patients, total TB-related patient costs averaged US$960 per patient, compared to an average total cost of US$6880 for 14 participating MDR-TB patients.	STRONG
Marks et al. [84]	2005–2007	USA	98	Population-based study	Average hospitalization cost per XDR-TB patient (US$285,000) was 3.5 times that per MDR-TB patient (US$81,000), in 2010 dollars	STRONG
Marks et al. [85]	2005–2007	USA	135	Retrospective study	Direct costs, mostly covered by the public sector, averaged $134,000 per MDR TB and $430,000 per XDR TB patient; in comparison, estimated cost per non-MDR TB patient is $17,000.	STRONG
HIV						
Krentz et al. [86]	2007–2011	Canada	NR	An observational cohort study	Patients with no resistance had mean PPPM costs of CDN 1058, by contrast with the CDN 1291 costs of patients with secondary/acquired resistance. Mean costs for one, two or three ARV class resistance was CDN 1278, 1337 and 1801, respectively.	STRONG
Martin S. [87]	2007	NR	NR	NR	Total annual costs ranged from $31 700 for a patient on a 1st-line highly active antiretroviral therapy (HAART) regimen to $42,600 on 6th-line.	WEAK
Meenan RT. [88]	2010	NR	NR	NR	Total mean healthcare costs were $35,000 higher for patients on a 3rd- or greater line treatment regimen compared to patients on a 1st- or 2nd-line treatment regimen.	WEAK

NR, not reported.

**Table 4 ijerph-17-01395-t004:** Strength of evidence by topic.

	Malaria	Tuberculosis	HIV Infection
Public health outcome	+++	+++	+
Clinical outcome	+++	+++	+++
Economic outcome	+	+++	++

+ reported in one moderate evidence article, or multiple weak evidence articles; ++ reported in one strong evidence article, or multiple moderate evidence articles; +++ reported in multiple strong evidence articles.

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
