# Peer review of "Outcome Impacts Due to Pathogen-Specific Antimicrobial Resistance: A Narrative Review of Published Literature"

_ijerph, 2020, doi:10.3390/ijerph17041395_

Round 1
Reviewer 1 Report
The manuscript presents a review of the public health, clinical, and economic impacts of antimicrobial resistance with a focus on three diseases: malaria, tuberculosis, and HIV. This topic is highly relevant considering the increasing recognition of AMR as a global public health threat and the narrative synthesis offers a clear summary of the literature. However, it is unclear why the authors did not follow PRISMA guidelines for systematic (or scoping) reviews and my main concerns/suggestions about this manuscript are as follows:
Search strategy: the described keywords used for the search are very limited and MeSH terms are not usable on Google Scholar. The search should be described in such a way that it can be repeated in the future. The keywords used on the two searched databases should be listed (in a table or supplementary materials). The criteria used at the different screening steps (title, abstract, full text) should also be described in more detail; given that modelling studies were included in the review, what is meant by “empirical” should be clarified. Definitions for public health, clinical, and public health outcomes considered in the review should also be provided for more clarity.
Quality/risk of bias assessment: an assessment of the quality of evidence from included studies would strengthen the manuscript overall and would be helpful to orient recommendations and future research.
Results presentation: Search results should be reported, i.e. # references retrieved in total, # passing title, abstract, and full text screening – for example a PRSIMA flow chart. Providing both n and % studies on each of the three categories and diseases considered in the review would be helpful. Specifying the key outcomes (and related findings?) for each study in Tables 1-3 would be helpful. Quality/risk of bias assessment results should also be included if available. Use citation style from the manuscript so that studies can easily be identified in references (i.e. reference number instead of year, or both). Detailed descriptions of individual studies have limited value on their own. The synthesis would benefit from summary sentences at the end of each § to highlight key trends/findings across studies. If relevant, specific research gaps could also be identified. The quantity (# studies) and quality of evidence available on each type of outcome and disease – for which ones have impacts been clearly established (and what are these impacts), what areas of further research needs? – could be synthesized in a table to provide an overview of the review results.
Discussion: Questions that would be worth addressing in the discussion include: How do the three diseases compare in terms of expected AMR impacts? Have the disease agents been developing AMR at similar rates? Are there disease-specific challenges in terms of outbreak control, clinical treatment & costs, research, …? Limitations of the presented work and selected approach are currently not acknowledged. For example, what are the types of documents that may have been missed by the search, and that could inform this work (e.g. outbreak reports?)? It is unclear why the end of the discussion focuses on gonorrhoea, when the review results do not include any study on that disease. How many studies were identified through the review linking AMR and gonorrhoea?
A few minor points are noted below.
Introduction
Line 31-32: reference for the definition of AMR.
Line 49: “early” instead of “easy”
Results
Line 93: “several reports…” – specify how many.
Lines 101-102: units are unclear – 10% of what? Relative risk of 5.9 comparing which groups?
Line 192: ref. 69 is a meta-analysis, thus not an empirical study – inclusion criteria need to be clarified.
Reviewer 2 Report
The standard methodology for review work such as PRISMA or Cochrane standard is recommended. This purpose is to avoid "risk of biased assessment and demonstrated the steps taken in selection of studies in line general acceptable format.
Line 31-32: There is WHO reference for this definition of AMR. Cite the reference or use cite another study where the definition was used and WHO reference was cited.
Result: Inclusion of a figure with summary of the studies used is desirable. Bar chart will be useful to demonstrate the number of studies with diseases and outcomes on a single figure.
Round 2
Reviewer 2 Report
Thanks for considering the previous suggestions during first review.